# Population modeling of tumor growth curves and the reduced Gompertz model improve prediction of the age of experimental tumors

**Cristina Vaghi**[1,2], **Anne Rodallec**[3], **Raphaëlle Fanciullino**[3], **Joseph Ciccolini**[3], **Jonathan P. Mochel**[4], **Michalis Mastri**[5], **Clair Poignard**[1,2], **John M. L. Ebos**[5,6], **Sébastien Benzekry**[1,2]*

**1** MONC team, Inria Bordeaux Sud-Ouest, Talence, France, **2** Institut de Mathématiques de Bordeaux, CNRS UMR 5251, Bordeaux University, Talence, France, **3** SMARTc Unit, Centre de Recherche en Cancérologie de Marseille, Inserm U1068, Aix Marseille Université, Marseille, France; Laboratoire de Pharmacocinétique et Toxicologie, La Timone University Hospital of Marseille, Marseille, France, **4** Department of Biomedical Sciences, College of Veterinary Medicine, Iowa State University, Ames, Iowa, United States of America, **5** Department of Cancer Genetics and Genomics, Roswell Park Comprehensive Cancer Center, Buffalo, New York, United States of America, **6** Departments of Medicine and Experimental Therapeutics, Roswell Park Comprehensive Cancer Center, Buffalo, New York, United States of America

* sebastien.benzekry@inria.fr

**Data Availability Statement:** The data are publicly accessible at the following repositories:

## Abstract

Tumor growth curves are classically modeled by means of ordinary differential equations. In analyzing the Gompertz model several studies have reported a striking correlation between the two parameters of the model, which could be used to reduce the dimensionality and improve predictive power. We analyzed tumor growth kinetics within the statistical framework of nonlinear mixed-effects (population approach). This allowed the simultaneous modeling of tumor dynamics and inter-animal variability. Experimental data comprised three animal models of breast and lung cancers, with 833 measurements in 94 animals. Candidate models of tumor growth included the exponential, logistic and Gompertz models. The exponential and—more notably—logistic models failed to describe the experimental data whereas the Gompertz model generated very good fits. The previously reported population-level correlation between the Gompertz parameters was further confirmed in our analysis ($R^2 > 0.92$ in all groups). Combining this structural correlation with rigorous population parameter estimation, we propose a reduced Gompertz function consisting of a single individual parameter (and one population parameter). Leveraging the population approach using Bayesian inference, we estimated times of tumor initiation using three late measurement timepoints. The reduced Gompertz model was found to exhibit the best results, with drastic improvements when using Bayesian inference as compared to likelihood maximization alone, for both accuracy and precision. Specifically, mean accuracy (prediction error) was 12.2% versus 78% and mean precision (width of the 95% prediction interval) was 15.6 days versus 210 days, for the breast cancer cell line. These results demonstrate the superior predictive power of the reduced Gompertz model, especially when combined with Bayesian estimation. They offer possible clinical perspectives for personalized prediction of

https://zenodo.org/record/3574531, https://zenodo.org/record/3593919 and https://zenodo.org/record/3572401.

**Funding:** The authors received no specific funding for this work.

**Competing interests:** The authors have declared that no competing interests exist.

the age of a tumor from limited data at diagnosis. The code and data used in our analysis are publicly available at https://github.com/cristinavaghi/plumky.

## Author summary

Mathematical models for tumor growth kinetics have been widely used since several decades but mostly fitted to individual or average growth curves. Here we compared three classical models (exponential, logistic and Gompertz) using a population approach, which accounts for inter-animal variability. The exponential and the logistic models failed to fit the experimental data while the Gompertz model showed excellent descriptive power. Moreover, the strong correlation between the two parameters of the Gompertz equation motivated a simplification of the model, the reduced Gompertz model, with a single individual parameter and equal descriptive power. Combining the mixed-effects approach with Bayesian inference, we predicted the age of individual tumors with only few late measurements. Thanks to its simplicity, the reduced Gompertz model showed superior predictive power. Although our method remains to be extended to clinical data, these results are promising for the personalized estimation of the age of a tumor from limited measurements at diagnosis.

## Introduction

In the era of personalized oncology, mathematical modeling is a valuable tool for quantitative description of physiopathological phenomena [1, 2]. It allows for a better understanding of biological processes and generates useful individual clinical predictions, for instance for personalized dose adaptation in cancer therapeutic menagement [3]. Tumor growth kinetics have been studied since several decades both clinically [4] and experimentally [5]. One of the main findings of these early studies is that tumor growth is not entirely exponential, provided it is observed over a long timeframe (100 to 1000 folds of increase) [6]. The specific growth rate slows down and this deceleration can be particularly well captured by the Gompertz model [7, 6, 8]:

$$V(t) = V_{\text{inj}} e^{\frac{\alpha}{\beta}\left(1 - e^{-\beta t}\right)}, \tag{1}$$

where $V_{\text{inj}}$ is the initial tumor size at $t_{\text{inj}} = 0$ and $\alpha$ and $\beta$ are two parameters.

While the etiology of the Gompertz model has been long debated [9], several independent studies have reported a strong and significant correlation between the parameters $\alpha$ and $\beta$ in either experimental systems [6, 10, 11], or human data [11, 12, 13]. While some authors suggested this would imply a constant maximal tumor size (given by $V_{\text{inj}} e^{\frac{\alpha}{\beta}}$ in (1)) across tumor types within a given species [11], others argued that because of the presence of the exponential function, this so called 'carrying capacity' could vary over several orders of magnitude [14].

Mathematical models for tumor growth have been previously studied and compared at the level of individual kinetics and for prediction of future tumor growth [15, 16]. However, detailed studies of statistical properties of tumor growth models using a population approach (i.e. integrating structural dynamics with inter-subject variability [17]) are rare [18]. Nonlinear mixed effects modeling of the Gompertz model has been applied to several fields in biology, e.g. to model growth in Japanese quails [19] or broiler chicken growth [20]. In the field of

tumor growth modeling, studies using a population approach have mostly been conducted for perturbed tumor growth under the action of therapeutics (see e.g. [21] for a clinical study and [22] for a review). In a previous publication, our group has used a mixed-effects framework to compare the descriptive power of several unperturbed tumor growth models, yet without reporting visual predictive checks, analysis of residuals nor values of the population parameters (typical values and standard deviations of the random effects) [15]. Other related works include the coupling of tumor growth models with metastatic spreading [23, 24], or an analysis of tumor growth kinetics from different cell lines using the Simeoni model only [25, 18]. A calibrated model of lymphoma tumor growth has also been introduced and used for predictions in [26]. More complex mechanistic models have been proposed to investigate the link between biological processes and tumor growth dynamics and perform predictions, including angiogenesis [27] and solid stress [28]. A model for tumor-immune interactions has been developed and validated in [29, 30], demonstrating its ability to predict future prostate specific antigen dynamics based on several pre- and post-treatment initiation data points. Mathematical models of tumor growth inhibition were presented to assess tumor size dynamics in colorectal cancer [31] and adult diffuse low-grade gliomas [32]. Spatial models have also been widely proposed in a theoretical context but few of them have been compared to data (see [33] for an example on thyroidal lung nodules and [34, 35] for gliomas).

Here we provide a detailed and comparative analysis of statistical properties of multiple classical tumor growth models within a population framework, applied to a data set of 94 animals, including three animal models and two methods of tumor size quantification (versus 54 animals in [15]). The main focus and novelty of the work reported here is to analyze the above-mentioned correlation between Gompertz parameters using a population approach, in order to improve model-derived predictions. This led us to a simplified model with only one subject-specific parameter (and one population-specific), the "reduced Gompertz" model [11].

Using population distributions as priors allows to make predictions on new subjects by means of Bayesian algorithms [36, 37, 38]. The added value of the latter method is that only few measurements per individual are necessary to obtain reliable predictions. In contrast with previous work focusing on the *forward* prediction of the size of a tumor [15], the present study addresses the *backward* problem, i.e. the estimation of the age of a tumor [39]. This question is of fundamental importance in the clinic since the age of a tumor can be used as a proxy for determination of the invisible metastatic burden at diagnosis [24]. In turn, this estimation has critical implications for decision of the extent of adjuvant therapy [40]. Since predictions of the initiation time of clinical tumors are hardly possible to verify for clinical cases, we developed and validated our method using experimental data from multiple data sets in several animal models. This setting allowed to have enough measurements, on a large enough time frame in order to assess the predictive power of the methods.

## Materials and methods

The python code and the data used in our analysis are available at https://github.com/cristinavaghi/plumky.

### Ethics statement

Animal tumor model studies were performed in strict accordance with the recommendations in the Guide for the Care and Use of Laboratory Animals of the National Institutes of Health. Protocols used were approved by the Institutional Animal Care and Use Committee (IACUC) at Tufts University School of Medicine for studies using murine Lewis lung carcinoma (LLC) cells (Protocol: #P11-324) and at Roswell Park Cancer Institute (RPCI) for studies using

human LM2-4$^{LUC+}$ breast carcinoma cells (Protocol: 1227M). Institutions are AAALAC accredited and every effort was made to minimize animal distress [15].

For the breast data measured by fluorescence, guidelines for animal welfare in experimental oncology as recommended by European regulations (decree 2013-118 of February 1, 2013) were followed. All animal experiments were approved by the Animal Ethic Committee of the Aix-Marseille Université (CE14). The protocol was registered as #2017031717108767 at the French Ministry of Research. Mice were monitored daily for signs of distress, pain, decreased physical activity, or any behavioral change and weighted thrice a week. Water was supplemented with paracetamol (80 mg/kg/day) to prevent any metastasis-related pain [41].

## Mice experiments

The experimental data comprised three data sets. Animal tumor model studies were performed in strict accordance with guidelines for animal welfare in experimental oncology and were approved by local ethics committees. Precise description of experimental protocols was reported elsewhere (see [15] for the volume measurements and [41] for the fluorescence measurements).

**Breast data measured by volume (N = 66).**   This dataset is publicly available at the following repository [42]. It consisted of human LM2-4$^{LUC+}$ triple negative breast carcinoma cells originally derived from MDA-MB-231 cells. Animal studies were performed as described previously under Roswell Park Comprehensive Cancer Center (RPCCC) Institutional Animal Care and Use Committee (IACUC) protocol number 1227M [15, 24]. Briefly, animals were orthotopically implanted with LM2-4$^{LUC+}$ cells ($10^6$ cells at injection) into the right inguinal mammary fat pads of 6- to 8-week-old female severe combined immunodeficient (SCID) mice. Tumor size was measured regularly with calipers to a maximum volume of 2 cm$^3$, calculated by the formula $V = \pi/6 w^2 L$ (ellipsoid) where $L$ is the largest and $w$ is the smallest tumor diameter. The data were pooled from eight experiments conducted with a total of 581 observations. All LM2-4$^{LUC+}$ implanted animals used in this study are vehicle-treated animals from published studies [15, 24]. Vehicle formulation was carboxymethylcellulose sodium (USP, 0.5% w/v), NaCl (USP, 1.8% w/v), Tween-80 (NF, 0.4% w/v), benzyl alcohol (NF, 0.9% w/v), and reverse osmosis deionized water (added to final volume) and adjusted to pH 6 (see [43]) and was given at 10ml/kg/day for 7-14 days prior tumor resection.

**Breast data measured by fluorescence (N = 8).**   This dataset is publicly available at the following repository [44]. It consisted of human MDA-MB-231 cells stably transfected with dTomato lentivirus. Animals were orthotopically implanted (80,000 cells at injection) into the mammary fat pads of 6-week-old female nude mice. Tumor size was monitored regularly with fluorescence imaging. The data comprised a total of 64 observations. To recover the fluorescence value corresponding to the injected cells, we computed the ratio between the fluorescence signal and the volume measured in mm$^3$. We used linear regression considering the volume data of a different data set with same experimental setup (mice, tumor type and number of injected cells). The estimated ratio was $1.52 \cdot 10^9$ photons/(s $\cdot$ mm$^3$) with relative standard error of 11.3%, therefore the initial fluorescence signal was $1.22 \cdot 10^7$ photons/s.

**Lung data measured by volume (N = 20).**   This dataset is publicly available at the following repository [45]. It consisted of murine Lewis lung carcinoma cells originally derived from a spontaneous tumor in a C57BL/6 mouse [46]. Animals were implanted subcutaneously ($10^6$ cells at injection) on the caudal half of the back in anesthetized 6- to 8-week-old C57BL/6 mice. Tumor size was measured as described for the breast data to a maximum volume of 1.5 cm$^3$. The data was pooled from two experiments with a total of 188 observations.

## Tumor growth models

We denote by $t_I$ and $V_I$ the initial conditions of the equation. At time of injection ($t = 0$), we assumed that all tumors within a group had the same size/volume $V_{inj}$ (equal to the number of injected cells converted into the appropriate unit) and denoted by $\alpha$ the specific growth rate (i.e. $\frac{1}{V}\frac{dV}{dt}$) at this time and size.

We considered the exponential, logistic and Gompertz models [15]. The first two are respectively defined by the following equations

$$\begin{cases} \dfrac{dV}{dt} = \alpha V, \\ V(t_I) = V_I, \end{cases} \quad \text{and} \quad \begin{cases} \dfrac{dV}{dt} = \rho\left(1 - \dfrac{V}{K}\right)V, \\ V(t_I) = V_I. \end{cases} \tag{2}$$

In the logistic equation, $K$ is a carrying capacity parameter. It expresses a maximal reachable size due to competition between the cells (e.g. for space or nutrients). The quantity $\rho = \alpha\left(\frac{K}{K - V_{inj}}\right)$ is a coefficient related to the growth rate. For small values of $V_{inj}$, $\rho$ tends to $\alpha$.

The Gompertz model is characterized by an exponential decrease of the specific growth rate with rate denoted here by $\beta$. Although multiple expressions and parameterizations coexist in the literature, the definition we adopted here reads as follows:

$$\begin{cases} \dfrac{dV}{dt} = \left(\alpha - \beta\log\left(\dfrac{V}{V_{inj}}\right)\right)V, \\ V(t_I) = V_I. \end{cases} \tag{3}$$

Note that the injected volume $V_{inj}$ appears in the differential equation defining $V$. This is a natural consequence of our assumption of $\alpha$ as being the specific growth rate at $V = V_{inj}$. This model exhibits sigmoidal growth up to a saturating value given by $K = V_{inj}e^{\frac{\alpha}{\beta}}$. Note also that the value of $K$ in the Gompertz model is independent of the initial data ($t_I, V_I$). The latter was considered to be $(0, V_{inj})$ when performing population analysis, while it was set equal to the observation $y^i_{n^i-2}$ of an animal $i$ for backward prediction (see section Individual predictions).

## Population approach

Let $N$ be the number of subjects within a population (group) and $\mathbf{Y}^i = \{y^i_1, ..., y^i_{n^i}\}$ the vector of longitudinal measurements in animal $i$, where $y^i_j$ is the observation of subject $i$ at time $t^i_j$ for $i = 1, ..., N$ and $j = 1, ..., n^i$ ($n^i$ is the number of measurements of individual $i$). We assumed the following observation model

$$y^i_j = f(t^i_j; \boldsymbol{\theta}^i) + e^i_j, \qquad j = 1, ..., n^i, \quad i = 1, ..., N, \tag{4}$$

where $f(t^i_j; \boldsymbol{\theta}^i)$ is the evaluation of the tumor growth model at time $t^i_j$, $\boldsymbol{\theta}^i \in \mathbb{R}^p$ is the vector of the parameters relative to the individual $i$ and $e^i_j$ the residual error model, to be defined later. An individual parameter vector $\boldsymbol{\theta}^i$ depends on fixed effects $\boldsymbol{\mu}$, identical within the population, and on a random effect $\boldsymbol{\eta}^i$, specific to each animal. Random effects follow a normal distribution with mean zero and variance matrix $\boldsymbol{\omega}$. Specifically:

$$\log(\boldsymbol{\theta}^i) = \log(\boldsymbol{\mu}) + \boldsymbol{\eta}^i, \quad \boldsymbol{\eta}^i \sim \mathcal{N}(0, \boldsymbol{\omega}).$$

The choice of a log-normal distribution ensured the positivity of the parameters without adding any constraint. Moreover, the ratio of two log-normal distributions is a log-normal distribution.

We considered a combined residual error model $e_j^i$, defined as

$$e_j^i = (\sigma_1 + \sigma_2 f(t_j^i; \boldsymbol{\theta}^i))\varepsilon_j^i,$$

where $\varepsilon_j^i \sim \mathcal{N}(0, 1)$ are the residual errors and $\boldsymbol{\sigma} = [\sigma_1, \sigma_2]$ is the vector of the residual error model parameters.

In order to compute the population parameters, we maximized the population likelihood, obtained by pooling all the data together. Usually, this likelihood cannot be computed explicitly for nonlinear mixed-effect models. We used the stochastic approximation expectation minimization algorithm (SAEM) [17], implemented in the `Monolix 2018 R2` software [47]. This algorithm is a variation of the EM algorithm, where the expectation step is replaced by a stochastic approximation of the likelihood function [48]. This method has been proven to efficiently converge to the maximum likelihood estimator for nonlinear mixed effects models [17].

In the remainder of the manuscript we will denote by $\phi = \{\boldsymbol{\mu}, \boldsymbol{\omega}, \boldsymbol{\sigma}\}$ the set of the population parameters containing the fixed effects $\boldsymbol{\mu}$, the covariance of the random effects $\boldsymbol{\omega}$ and the error model parameters $\boldsymbol{\sigma}$.

## Individual predictions

For a given animal $i$, the backward prediction problem we considered was to predict the age of the tumor based on the three last measurements $\boldsymbol{y}^i = \{y_{n^i-2}^i, y_{n^i-1}^i, y_{n^i}^i\}$. Since we were in an experimental setting, we considered the injection time as the initiation time and thus the age was given by $a^i = t_{n^i-2}^i$. Then, we considered as model $f(t; \boldsymbol{\theta}^i)$ the solution of the Cauchy problem (3) endowed with initial conditions $(t_I^i = t_{n^i-2}^i, V_I^i = y_{n^i-2}^i)$. For estimation of the parameters (estimate $\hat{\boldsymbol{\theta}}^i$), we applied two different methods: likelihood maximization alone (no use of prior population information) and Bayesian inference (use of prior). The predicted age $\hat{a}^i$ was then defined by

$$f(t_{n^i-2}^i - \hat{a}^i; \hat{\boldsymbol{\theta}}^i) = V_{\text{inj}},$$

that is:

$$\hat{a}^i = \frac{1}{\hat{\beta}^i}\left(\log\left(\frac{\hat{\alpha}^i}{\hat{\beta}^i}\right) - \log\left(\frac{\hat{\alpha}^i}{\hat{\beta}^i} - \log\left(\frac{V_I^i}{V_{\text{inj}}}\right)\right)\right) \tag{5}$$

in case of the Gompertz model.

**Likelihood maximization.** For individual predictions with likelihood maximization, no prior information on the distribution of the parameters was used. Parameters of the error model were not re-estimated: values from the population analysis were used. The log-

likelihood can be derived from (4):

$$
\begin{aligned}
l(\boldsymbol{\theta^i}) &= \ln\left(\prod_{j=n^i-2}^{n^i} \mathbb{P}(y_j^i|\boldsymbol{\theta^i})\right) \\
&= -\frac{3}{2}\log(2\pi) - \frac{1}{2}\sum_{j=n^i-2}^{n^i}\left(\log\left(\sigma_1 + \sigma_2 f\left(t_j^i, \boldsymbol{\theta^i}\right)\right) + \left(\frac{y_j^i - f(t_j^i, \boldsymbol{\theta^i})}{\sigma_1 + \sigma_2 f(t_j^i, \boldsymbol{\theta^i})}\right)^2\right),
\end{aligned}
\tag{6}
$$

where $\mathbb{P}(y_j^i|\boldsymbol{\theta^i})$ is the likelihood of the observation of the animal $i$ at time $t_j^i$.

In order to guarantee the positivity of the parameters, we introduced the relation $\boldsymbol{\theta^i} = g(\boldsymbol{\gamma^i}) = e^{\gamma^i}$ and substituted this in Eq (6). The negative of Eq (6) was minimized with respect to $\boldsymbol{\gamma^i}$ (yielding the maximum likelihood estimate $\hat{\boldsymbol{\gamma}}^i$) with the function `minimize` of the python module `scipy.optimize`, for which the Nelder-Mead algorithm was applied. Thanks to the invariance property, the maximum likelihood estimator of $\boldsymbol{\theta^i}$ was determined as $\hat{\boldsymbol{\theta}}^i = e^{\hat{\gamma}^i}$. Individual prediction intervals were computed by sampling the parameters $\boldsymbol{\theta^i}$ from a gaussian distribution with variance-covariance matrix of the estimate defined as $\nabla g(\hat{\boldsymbol{\gamma}}^i)^T \cdot (\hat{s}^{2,i}(\boldsymbol{I}^{-1}(\hat{\boldsymbol{\gamma}}^i))) \cdot \nabla g(\hat{\boldsymbol{\gamma}}^i)$ where $\hat{s}^{2,i} = \frac{1}{3-p}\sum_{j=n^i-2}^{n^i}\left(\frac{y_j^i - f(t_j^i,\hat{\boldsymbol{\theta}}^i)}{\sigma_1 + \sigma_2 f(t_j^i,\hat{\boldsymbol{\theta}}^i)}\right)^2$, with $p$ the number of parameters (and the factor 3 in the denominator because this is the number of observations), $\boldsymbol{I}(\hat{\boldsymbol{\gamma}}^i)$ the Fisher information matrix and $\nabla g(\hat{\boldsymbol{\gamma}}^i)$ the gradient of the function $g(\boldsymbol{\gamma})$ evaluated in the estimate $\hat{\boldsymbol{\gamma}}^i$. Denoting by $\boldsymbol{f}(\boldsymbol{\gamma}) = [f(t_j^i, e^\gamma)]_{j=n^i-2}^{n^i}$ and by $\boldsymbol{\Omega}(\boldsymbol{\gamma}) = \mathrm{diag}(\sigma_1 + \sigma_2[f(t_j^i, e^\gamma)]_{j=n^i-2}^{n^i})$, the Fisher information matrix was defined by [49]:

$$
[\boldsymbol{I}(\boldsymbol{\gamma})]_{l,m} = \left[\frac{\partial \boldsymbol{f}(\boldsymbol{\gamma})}{\partial \gamma_l}\right]^T \boldsymbol{\Omega}^{-1}(\boldsymbol{\gamma})\left[\frac{\partial \boldsymbol{f}(\boldsymbol{\gamma})}{\partial \gamma_m}\right] + \frac{1}{2}\mathrm{tr}\left[\boldsymbol{\Omega}^{-1}(\boldsymbol{\gamma})\frac{\partial \boldsymbol{\Omega}(\boldsymbol{\gamma})}{\partial \gamma_l}\boldsymbol{\Omega}^{-1}(\boldsymbol{\gamma})\frac{\partial \boldsymbol{\Omega}(\boldsymbol{\gamma})}{\partial \gamma_m}\right].
\tag{7}
$$

**Bayesian inference.** When applying the Bayesian method, we considered *training sets* to learn the distribution of the parameters $\phi$ and *test sets* to derive individual predictions. For a given animal $i$ of a *test set*, we predicted the age of the tumor based on the combination of: 1) population parameters $\phi$ identified on the *training set* using the population approach and 2) the three last measurements of animal $i$. We set as initial conditions $t_I = 0$ and $V_I^i \sim \mathcal{N}(y_{n^i-2}^i, \sigma_1 + \sigma_2 y_{n^i-2}^i)$. We considered the initial volume $V_I$ to be a random variable to account for measurement uncertainty on $y_{n^i-2}^i$. We then estimated the posterior distribution $\mathbb{P}(\boldsymbol{\theta^i}|\boldsymbol{y^i})$ of the parameters $\boldsymbol{\theta^i}$ using a Bayesian approach [37]:

$$
\mathbb{P}(\boldsymbol{\theta^i}|\boldsymbol{y^i}) = \frac{\mathbb{P}(\boldsymbol{y^i}|\boldsymbol{\theta^i})\mathbb{P}(\boldsymbol{\theta^i})}{\mathbb{P}(\boldsymbol{y^i})},
\tag{8}
$$

where $\mathbb{P}(\boldsymbol{\theta^i})$ is the prior distribution of the parameters estimated through nonlinear mixed-effects modeling (*i.e.*, the population parameters $\phi$), $\mathbb{P}(\boldsymbol{y^i}|\boldsymbol{\theta^i}) = \int_\mathbb{R}\mathbb{P}(V_I^i)\mathbb{P}(\boldsymbol{y^i}|\boldsymbol{\theta^i}, V_I^i)dV_I^i$ is the likelihood, defined from Eq (4), and $\mathbb{P}(\boldsymbol{y^i}) = \int_{\mathbb{R}^p}\mathbb{P}(\boldsymbol{\theta^i})\mathbb{P}(\boldsymbol{y^i}|\boldsymbol{\theta^i})d\boldsymbol{\theta^i}$ is a normalization factor. The predicted distributions of extrapolated growth curves and subsequent $\hat{a}^i$ were computed by sampling $\boldsymbol{\theta^i}$ from its posterior distribution (8) using `Pystan`, a Python interface to the software `Stan` [38] for Bayesian inference based on the No-U-Turn sampler, a variant of Hamiltonian Monte Carlo [36]. The sampling procedure depends on the evaluation of the likelihood $\mathbb{P}(\boldsymbol{y^i}|\boldsymbol{\theta^i})$, which relies itself on $V_I^i$. Therefore, $V_I^i$ was sampled from its distribution for each

realization of the posterior distribution. Predictions of $\hat{a}^i$ were then obtained from (5), considering the median value of the distribution.

Different data sets were used for learning the priors (*training sets*) and prediction (*test sets*) by means of *k*-fold cross validation, with *k* equal to the total number of animals of the dataset ($k = N$, i.e. leave-one-out strategy). At each iteration we computed the parameters distribution of the population composed by $N - 1$ individuals and used this as prior to predict the initiation time of the excluded subject *i*. The `Stan` software was used to draw 2000 realizations from the posterior distribution of the parameters of the individual *i*.

## Results

Results were similar for the three data sets presented in the materials and methods. For conciseness, the results presented below are related to the largest dataset (breast cancer data measured by volume). Results relative to the other datasets are reported in S1–S10 Figs and S1–S4 Tables.

### Population analysis of tumor growth curves

The population approach was applied to test the descriptive power of the exponential, logistic and Gompertz models for tumor growth kinetics. The number of injected cells at time $t_{inj} = 0$ was $10^6$, therefore we fixed the initial volume $V_{inj} = 1$ mm$^3$ in the whole dataset [15]. We set $(t_I, V_I) = (t_{inj}, V_{inj})$ as initial condition of the equations.

We ran the SAEM algorithm with the `Monolix` software to estimate the fixed and random effects [47]. Moreover, we evaluated different statistical indices in order to compare the different tumor growth models. This also allowed learning of the parameter population distributions that were used later as priors for individual predictions. Results are reported in Table 1, where the models are ranked according to their AIC (Akaike Information Criterion), a metrics combining parsimony and goodness-of-fit. The Gompertz model was the one with the lowest values, indicating superior goodness-of-fit. This was confirmed by diagnostic plots (Fig 1). The visual predictive checks (VPCs) in Fig 1A compare the empirical percentiles with the theoretical percentiles, i.e. those obtained from simulations of the calibrated models. The VPC of the exponential and logistic models showed clear model misspecification. On the other hand, the VPC of the Gompertz model was excellent, with observed percentiles close to the predicted ones and small prediction intervals (indicative of correct identifiability of the parameters). Fig 1B shows the prediction distributions of the three models. This allowed to compare the observations with the theoretical distribution of the predictions. Only the prediction distribution of the Gompertz model covered the entire dataset. The logistic model exhibited a saturation of tumor dynamics at lower values than compatible with the data.

Moreover, the distribution of the residuals was symmetrical around a mean value of zero with the Gompertz model (Fig 1C), strengthening its good descriptive power, while the

**Table 1. Models ranked in ascending order of AIC (Akaike information criterion).** Other statistical indices are the log-likelihood estimate (-2LL) and the Bayesian information criterion (BIC). *The reduced Gompertz model is introduced below.

| Model | -2LL | AIC | BIC |
|---|---|---|---|
| Gompertz | 7129 | 7143 | 7158 |
| Reduced Gompertz* | 7259 | 7269 | 7280 |
| Logistic | 7584 | 7596 | 7609 |
| Exponential | 8652 | 8660 | 8669 |

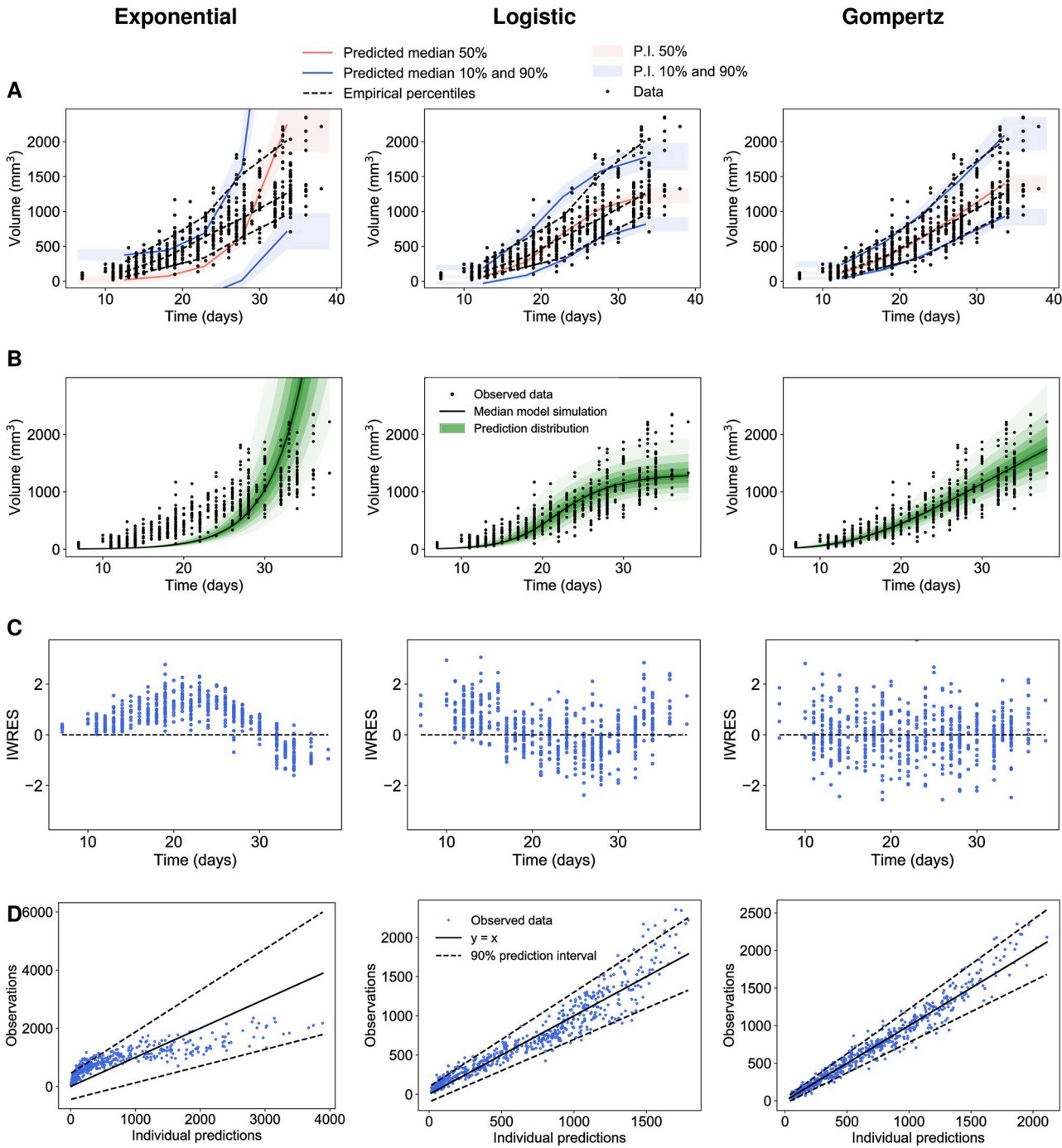

**Fig 1. Population analysis of experimental tumor growth kinetics.** (A) Visual predictive checks assess goodness-of-fit for both structural dynamics and inter-animal variability by reporting model-predicted percentiles (together with confidence prediction intervals (P.I) in comparison to empirical ones. They were obtained by multiple simulations of each model. The time axis was then split into bins and in each interval the empirical percentiles of the observed data were compared with the respective predicted medians and intervals of the simulated data [47]. (B) Prediction distributions. They were obtained by multiple simulations of all individuals in the dataset, excluding the residual error [47]. (C) Individual weighted residuals (IWRES) with respect to time. (D) Observations vs predictions Left: exponential, Center: logistic, Right: Gompertz models.

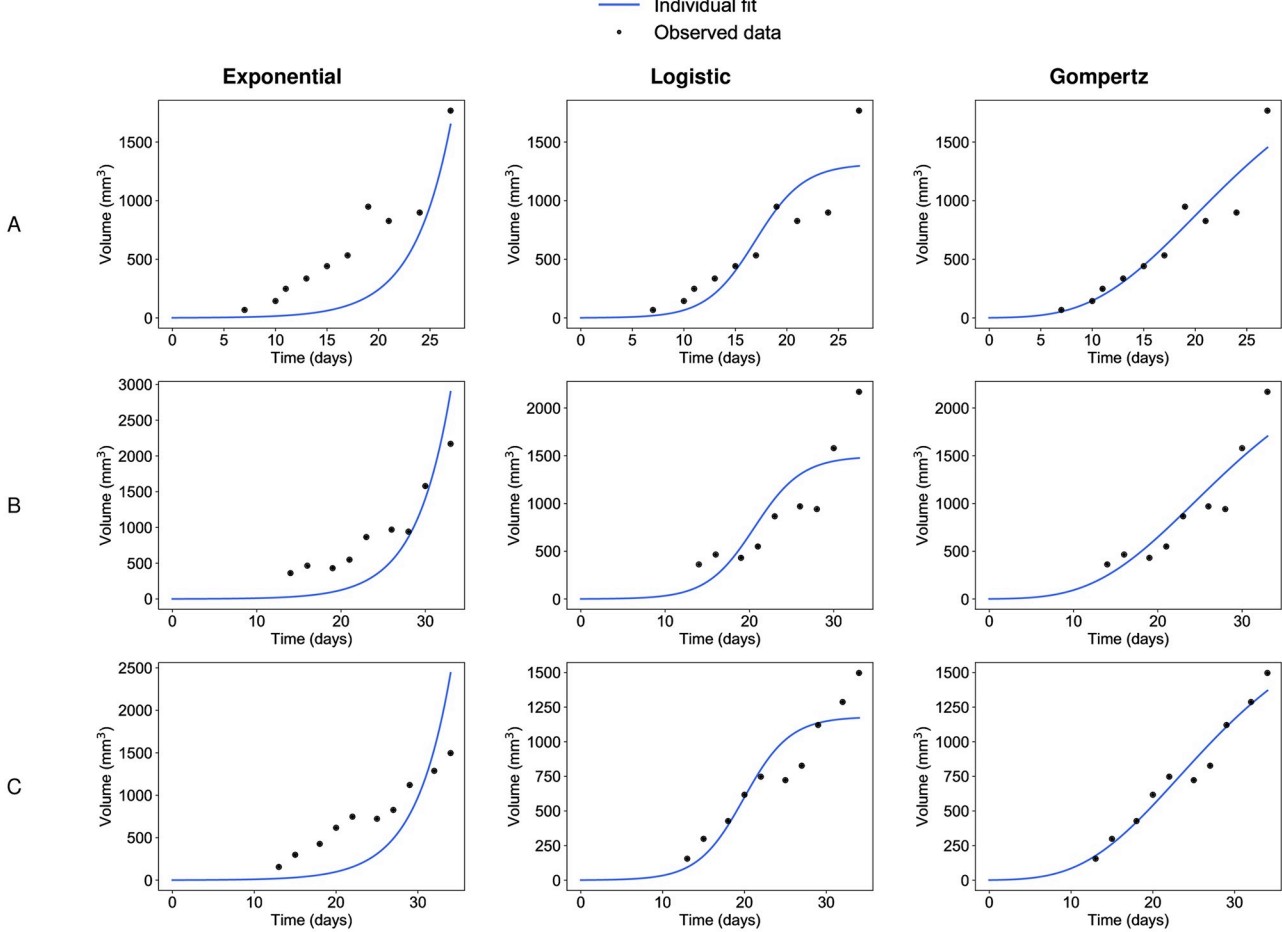

**Fig 2. Individual fits from population analysis.** Three representative examples of individual fits (animal (A), animal (B) and animal (C)) computed with the population approach relative to the exponential (left), the logistic (center) and the Gompertz (right) models.

exponential and logistic models exhibited clear skewed distributions. The observations vs individual predictions in Fig 1D further confirmed these findings.

These observations at the population level were confirmed by individual fits, computed from the mode of the posterior conditional parameter distribution for each individual (Fig 2). Confirming previous results [15], the optimal fits of the exponential and logistic models were unable to give appropriate description of the data, suggesting that these models should not be used to describe tumor growth, at least in similar settings to ours. Fitting of late timepoints data forced the proliferation parameter of the exponential model to converge towards a rather low estimate, preventing reliable description of the early datapoints. The converse occurred for the logistic. Constrained by the early data points imposing to the model the pace of the growth deceleration, the resulting estimation of the carrying capacity $K$ was biologically irrelevant (much too small, typical value 1303 mm$^3$, see Table 2), preventing the model to give a good description of the late growth.

Table 2 provides the values of the population parameters. The relative standard error estimates associated to population parameters were all rather low (<3.81%), indicating good practical identifiability of the model parameters. Standard error estimates of the constant error model parameters were found to be slightly larger (<19.3%), suggesting that for some models

**Table 2. Fixed effects (typical values) of the parameters of the different models.** Par. = parameter. $\omega$ = standard deviation of the random effects. R.S.E. = relative standard errors of the estimates. $\sigma$ = residual error model parameters. *The reduced Gompertz model is introduced below.

| Model | Par. | Unit | Fixed effects | $\omega$ | R.S.E. (%) |
|---|---|---|---|---|---|
| Gompertz | $\alpha$ | day$^{-1}$ | 0.58 | 0.19 | 2.51 |
| | $\beta$ | day$^{-1}$ | 0.072 | 0.26 | 3.42 |
| | $\sigma$ | - | [20.5, 0.11] | - | [16.9, 7.53] |
| Reduced Gompertz* | $\beta$ | day$^{-1}$ | 0.075 | 0.13 | 1.74 |
| | $k$ | - | 7.87 | - | 0.21 |
| | $\sigma$ | - | [14.8, 0.17] | - | [19.3, 5.32] |
| Logistic | $\rho$ | day$^{-1}$ | 0.325 | 0.138 | 1.82 |
| | $K$ | mm$^3$ | 1303 | 0.25 | 3.81 |
| | $\sigma$ | - | [58.9, 0.12] | - | [8.97, 9.14] |
| Exponential | $\alpha$ | day$^{-1}$ | 0.231 | 0.08 | 1.38 |
| | $\sigma$ | - | [272, 0.26] | - | [6.10, 15.1] |

a proportional error model might have been more appropriate—but not in case of the exponential model. Since our aim was to compare different tumor growth equations, we established a common error model parameter, i.e. a combined error model. Relative standard errors of the standard deviations of the random effects $\omega$ were all smaller than 9.6% (not shown).

These model findings in the breast cancer cell line were further validated with the other cell lines. For both the lung cancer and the fluorescence-breast cancer cell lines, the Gompertz model outperformed the other competing models (see S1 and S2 Tables for goodness-of-fit metrics, and S3 and S4 Tables for parameter values), as also shown by the diagnostic plots (S1 and S2 Figs). Individual plots confirmed these observations and are provided in S3 and S4 Figs. For the fluorescence-breast cancer cell line the constant part of the error model was found negligible and we used a proportional error model (i.e., we fixed $\sigma_1 = 0$). Value of $\sigma_2$ was found particularly high for the Exponential model (S4 Table), which resulted in inappropriate fits (S2 and S4 Figs), further supporting rejection of this model. Estimated inter-individual variability for the other models was found small. This was probably due to the small number of animals in the data set.

Together, these results confirmed that the exponential and logistic models are not appropriate models of tumor growth while the Gompertz model has excellent descriptive properties, for both goodness-of-fit and parameter identifiability purposes.

## The reduced Gompertz model

**Correlation between the Gompertz parameters.** During the estimation process of the Gompertz parameters, we found a high correlation between $\alpha$ and $\beta$ within the population. At the population level, the SAEM algorithm estimated a correlation of the random effects equal to 0.981. At the individual level, $\alpha^i$ and $\beta^i$ were also highly linearly correlated (Fig 3A, $R^2 = 0.968$), confirming previous results [6, 11, 10, 12, 50]. This motivated the reformulation of the alpha parameter as follows:

$$\alpha^i = k\beta^i + c, \tag{9}$$

where $k$ and $c$ are representing the slope and the intercept of the regression line, respectively. In our analysis we found $c$ to be small ($c = 0.14$), thus we further assumed this term to be negligible and fixed it to 0. This suggests $k$ as a constant of tumor growth within a given animal model with similar characteristics (note however that from (3), $k$ depends on $V_{\text{inj}}$) [11, 51]. In

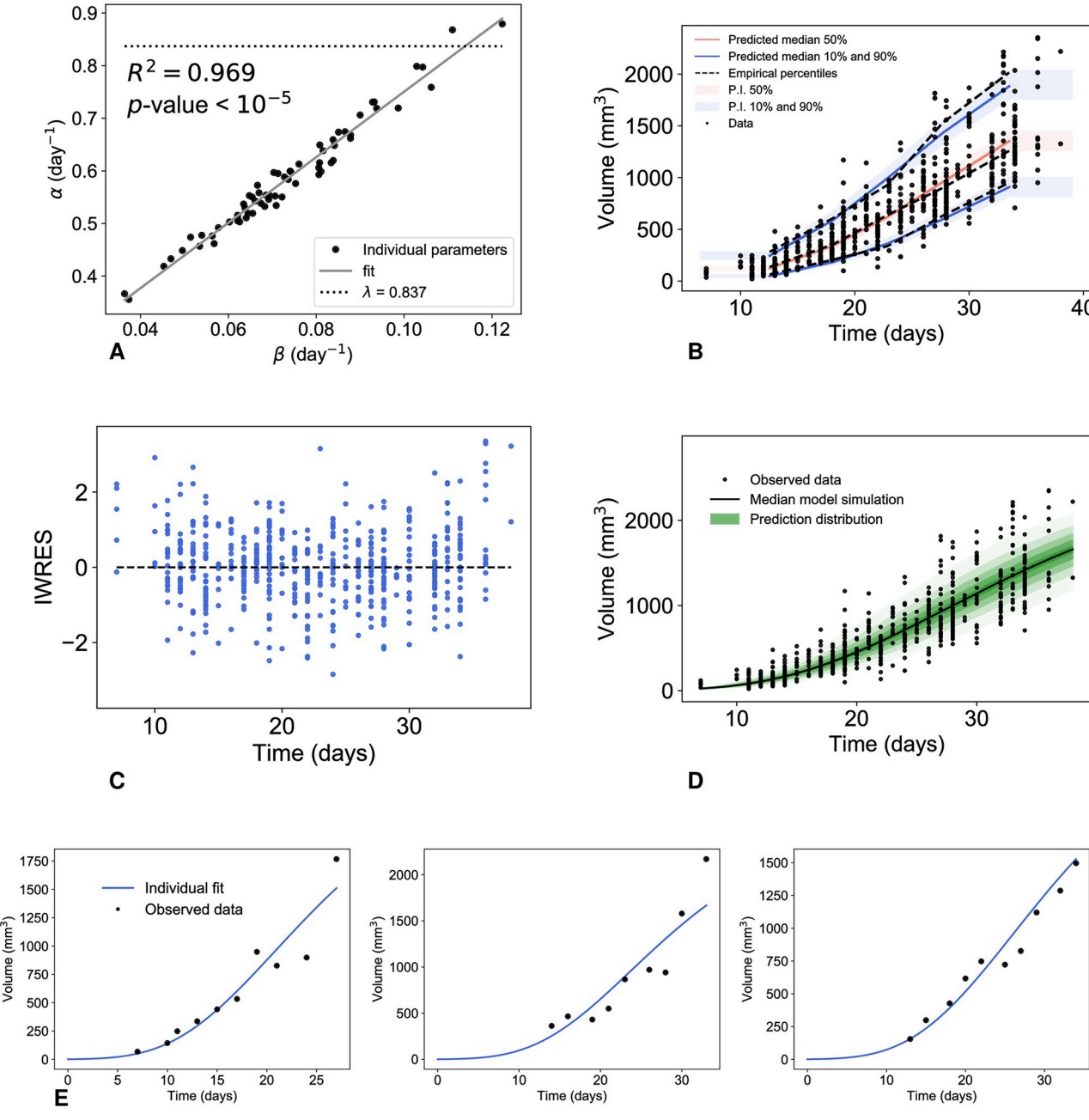

**Fig 3. Correlation of the Gompertz parameters and diagnostic plots of the reduced Gompertz model from population analysis.** Correlation between the individual parameters of the Gompertz model (A) and results of the population analysis of the reduced Gompertz model: visual predictive check (B), scatter plots of the residuals (C), prediction distribution (D) and examples of individual fits (E).

turn, this implies an approximately constant limiting size

$$K^i = V_{\text{inj}} e^{\frac{\alpha^i}{\beta^i}} \simeq V_{\text{inj}} e^k \simeq 2600 \text{ mm}^3, \quad \forall i.$$

The other data sets gave analogous results in terms of goodness of fit and correlation between $\alpha$ and $\beta$, even if the constant limiting size was found different in the three cell lines.

The estimated correlations of the random effects were 0.967 and 0.998 for the lung cancer and for the fluorescence-breast cancer, respectively. The correlation between the parameters was also confirmed at the individual level (see S5A and S6A Figs, $R^2$ was 0.923 and 0.99 for the two data sets, respectively).

**Biological interpretation in terms of the proliferation rate.** By definition, the parameter $\alpha^i$ is the specific growth rate (SGR) at the volume $V_{inj}$, simply assumed to be the volume corresponding to the number of injected cells within a given animal model (e.g. $V_{inj} = 1$ for the breast data measured by volume). Assuming that the cells don't change their proliferation kinetics when implanted, this value should thus in theory be equal to the *in vitro* proliferation rate (supposed to be the same for all the cells of the same cell line), denoted here by $\lambda$. The value of this biological parameter was assessed *in vitro* and estimated at 0.837 [24]. In support to our quantitative assumptions, we indeed found estimated values of $\alpha^i$ close to $\lambda$ (fixed effects of 0.58, see Table 2).

However, most of the values of $\alpha^i$ were smaller than $\lambda$ in the majority of the cases (Fig 3A). We postulated that this difference could be explained by the fact that not all the cells will be successfully grafted when injected in an animal. Under such assumption the SGR at the initial time, to be compared with $\lambda$, would not be given by $\alpha^i$ anymore. Instead, denoting by $\hat{V}^i_{inj} < V_{inj}$ the (unknown) volume of the successfully grafted cells, and assuming further that the SGR at initiation would be fixed and given by $\lambda$ leads to the following reformulation of the Gompertz model

$$\begin{cases} \dfrac{dV^i}{dt} = \left( \lambda - \beta^i \log\left( \dfrac{V^i}{\hat{V}^i_{inj}} \right) \right) V^i \\ V^i(t_I = 0) = \hat{V}^i_{inj} \end{cases}$$

In turn, fitting this model to the data provides estimates of the percentage of successful engraftment of 7% ± 12.5% (mean ± standard deviation).

Alternatively, these results might also be explained by a time lag between the cell implantation and the initiation of tumor growth, due to the time needed by the cells to adapt to the new environment [52]. However, the two interpretations are indistinguishable in our case and might require a more elaborate analysis with specific data.

**Population analysis of the reduced Gompertz model.** The high correlation among the Gompertz parameters, suggested that a reduction of the degrees of freedom (number of parameters) in the Gompertz model could improve identifiability and yield a more parsimonious model. We considered the expression (9), assuming $c$ to be negligible. We therefore propose the following reduced Gompertz model:

$$\begin{cases} \dfrac{dV^i}{dt} = \left( \beta^i k - \beta^i \log\left( \dfrac{V^i}{V_{inj}} \right) \right) V^i \\ V^i(t_I^i) = V_I^i \\ \log(\beta^i) = \log(\beta_{pop}) + \eta_\beta^i, \quad \eta_\beta^i \sim \mathcal{N}(0, \omega_\beta) \\ k = k_{pop} \end{cases} \qquad (10)$$

where $\beta$ has mixed effects, while $k$ has only fixed effects, i.e., is constant within the population.

Fig 3 shows the results relative to the population analysis of this reduced Gompertz model. Results of the diagnostic plots indicated no deterioration of the goodness-of-fit as compared

with the Gompertz model (Fig 3B–3D). Only on the last timepoint was the model slightly underestimating the data (Fig 3D), which might explain why the model performs slightly worse than the two-parameters Gompertz model in terms of strictly quantitative statistical indices (but still better than the logistic or exponential models, Table 1). Individual dynamics were also accurately described (Fig 3E). Parameter identifiability was also excellent (Table 2).

The other two data sets gave similar results (see S5 and S6 Figs).

Together, these results demonstrated the accuracy of the reduced Gompertz model, with improved robustness as compared to previous models.

## Prediction of the age of a tumor

Considering the increased robustness of the reduced Gompertz model (one individual parameter less than the Gompertz model), we further investigated its potential for improvement of predictive power. We considered the problem of estimating the age of a tumor, that is, the time elapsed between initiation (here the time of injection) and detection occurring at larger tumor size (Fig 4). For a given animal $i$, we considered as first observation $y^i_{n^i-2}$ and aimed to predict its age $a^i = t^i_{n^i-2}$ (see Methods). We compared the results given by the Bayesian inference with the ones computed with standard likelihood maximization method (see Methods). To that end, we did not consider any information on the distribution of the parameters. For the reduced Gompertz model however (likelihood maximization case), we used the value of $k$ calculated in the previous section (Table 2), thus using information on the entire population. Importantly, for both prediction approaches, our methods allowed not only to generate a prediction of $a^i$ for estimation of the model accuracy (i.e. absolute relative error of prediction), but also to estimate the uncertainty of the predictions (i.e. precision, measured by the width of the 95% prediction interval (PI)).

Fig 4 presents a few examples of prediction of three individuals without (LM) and with (Bayesian inference) priors relative to the breast cancer measured by volume. The reduced Gompertz model combined to Bayesian inference (bottom row) was found to have the best accuracy in predicting the initiation time (mean error = 12.2%, 8.8% and 12.3% for the volume-breast cancer, lung cancer and fluorescence-breast cancer respectively) and to have the smallest uncertainty (precision = 15.6, 7.79 and 23.6 days for the three data sets, respectively). Table 3 gathers results of accuracy and precision for the Gompertz and reduced Gompertz models under LM and Bayesian inference relative to the three data sets. With only local information of the three last data points, the Gompertz model predictions were very inaccurate (mean error = 156%, 178% and 236%) and the Fisher information matrix was often singular, preventing standard errors to be adequately computed. With one degree of freedom less, the reduced Gompertz model had better performances with LM estimation but still large uncertainty (mean precision under LM = 210, 103 and 368 days) and poor accuracy using LM (mean error = 79%, 68.9% and 91.7%). Examples shown in Fig 4 were representative of the entire population relative to the breast cancer measured by volume. Eventually, for 97%, 95% and 87.5% of the individuals of the three data sets the actual value of the age fell in the respective prediction interval when Bayesian inference was applied in combination with the reduced Gompertz models. This means a good coverage of the prediction interval and indicates that our precision estimates were correct. On the other hand, this observation was not valid in case of likelihood maximization, where the actual value fell in the respective prediction interval for only 42.4%, 35% and 75% of the animals when the reduced Gompertz model was used.

Addition of *a priori* population information by means of Bayesian estimation resulted in drastic improvement of the prediction performances (Fig 5). This result was confirmed in the the other data sets (see S7 and S8 Figs for the lung cell line and S9 and S10 Figs for the breast

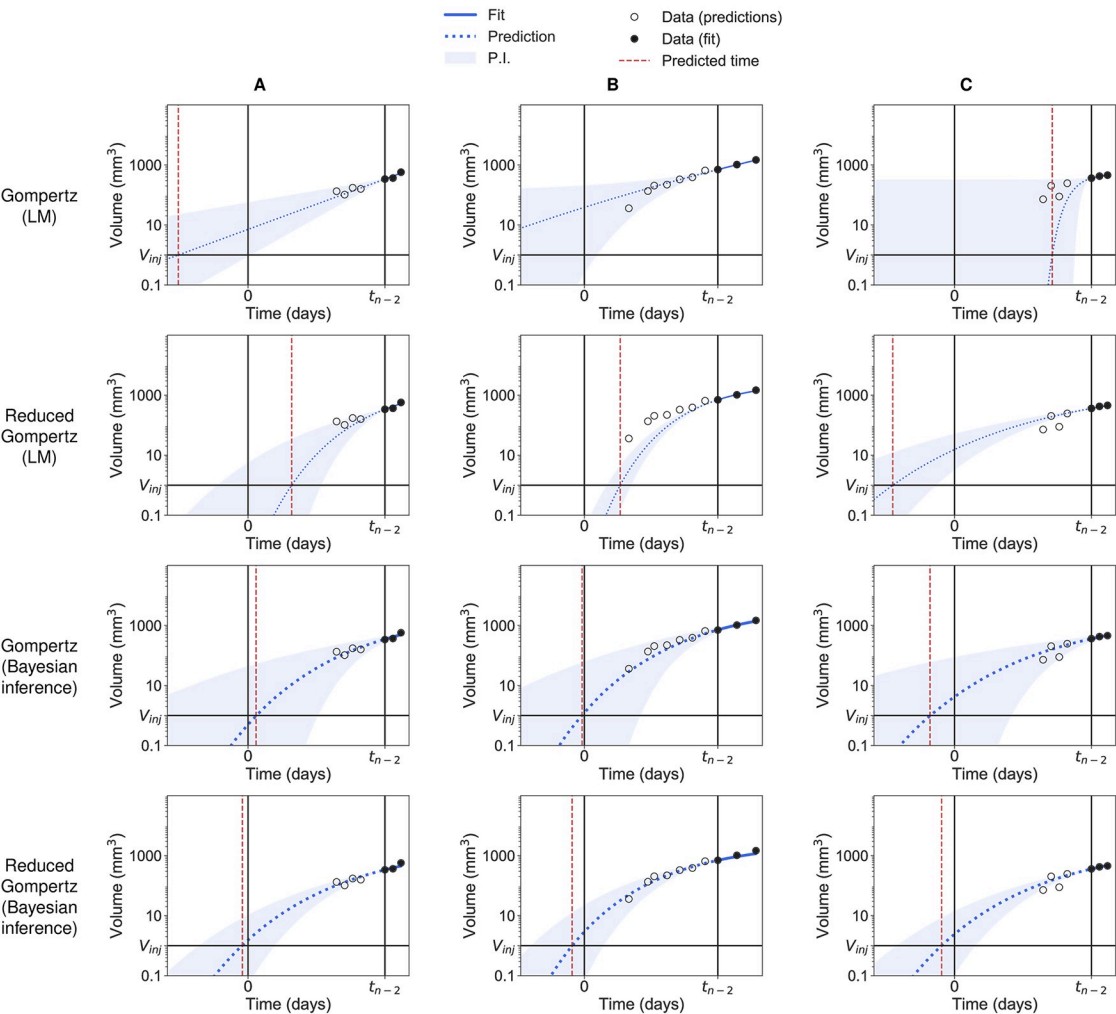

**Fig 4. Backward predictions computed with likelihood maximization and with Bayesian inference.** Examples of backward predictions of three individuals (A), (B) and (C) computed with likelihood maximization (LM) and Bayesian inference: Gompertz model with likelihood maximization (first row); reduced Gompertz with likelihood maximization (second row); Gompertz with Bayesian inference (third row) and reduced Gompertz with Bayesian inference (fourth row). Only the last three points are considered to estimate the parameters. The grey area is the 95% prediction interval (P.I) and the dotted blue line is the median of the posterior predictive distribution. The red line is the predicted initiation time and the black vertical line the actual initiation time.

cell line measured by fluorescence). For the breast and lung cancer cell lines measured by volume, a Wilcoxon test was performed to analyze the different error distributions shown in Figures Fig 5C and S8C Fig. For the fluorescence-breast cancer cell line we could not report a significant difference in terms of accuracy between the Gompertz and the reduced Gompertz when applying Bayesian inference. This can be explained by the low number of individuals included in the data set.

Overall, the combination of the reduced Gompertz model with Bayesian inference clearly outperformed the other methods for prediction of the age of experimental tumors.

## Discussion

We have analyzed tumor growth curves from multiple animal models and experimental techniques, using a population framework. This approach is ideally suited for experimental or

**Table 3. Accuracy and precision of methods for prediction of the age of experimental tumors of the three cell lines.** Accuracy was defined as the absolute value of the relative error (in percent). Precision was defined as the width of the 95% prediction interval (PI column, in days). Reported are the means and standard errors (in parenthesis). LM = likelihood maximization.

| Cell line | Model | Estimation method | Error | PI |
|---|---|---|---|---|
| Breast, volume | Reduced Gompertz | Bayesian | 12.2 (1.05) | 15.6 (0.509) |
| | Reduced Gompertz | LM | 79 (13.2) | 210 (58.6) |
| | Gompertz | Bayesian | 16.4 (1.65) | 41.1 (1.63) |
| | Gompertz | LM | 156 (21.7) | - |
| Lung, volume | Reduced Gompertz | Bayesian | 8.78 (1.43) | 7.79 (0.275) |
| | Reduced Gompertz | LM | 68.9 (33.1) | 103 (92.6) |
| | Gompertz | Bayesian | 18.9 (2.87) | 19.7 (1.89) |
| | Gompertz | LM | 178 (71.6) | - |
| Breast, fluorescence | Reduced Gompertz | Bayesian | 12.3 (2.9) | 23.6 (5.15) |
| | Reduced Gompertz | LM | 91.7 (21.1) | 368 (223) |
| | Gompertz | Bayesian | 13.5 (3.5) | 45.4 (4.43) |
| | Gompertz | LM | 236 (150) | - |

clinical data of the same tumor type within a given group of subjects. Indeed, it allows for a description of the inter-subject variability that is impossible to obtain when fitting models to averaged data (as often done for tumor growth kinetics [53]), while enabling a robust population-level description that is strictly more informative than individual fits alone. As expected from the classical observation of decreasing specific growth rates [6, 54, 8, 55, 56], the exponential model generated very poor fits. More surprisingly given its popularity in the theoretical community (probably due to its ecological ground), the logistic model was also rejected, due to unrealistically small inferred value of the carrying capacity $K$. This finding confirms at the population level previous results obtained from individual fits [15, 57]. It suggests that the underlying theory (competition between the tumor cells for space or nutrients) is unable—at least when considered alone—to explain the decrease of the specific growth rate, suggesting that additional mechanisms need to be accounted for. Indeed, the logistic model relies on space-independent cellular interactions, which might be biologically unrealistic [58]. Few studies have previously compared the descriptive performances of growth models on the same data sets [15, 59, 16]. In contrast to our results, Vaidya and Alexandro [16] found admissible description of tumor growth data employing the logistic model. Beyond the difference of animal model, we believe that the major reason explaining this discrepancy is the type of error model that was employed, as also noticed by others [57]. Here we used a combined error model, in accordance to our previous study [15] that had examined repeated measurements of tumor size and concluded to rejection of a constant error model (used in [16]). Moreover, statistical goodness-of-fit metrics were substantially worse when using a constant error model (e.g AIC of 7362 versus 7129, for the Gompertz model, results not shown). To avoid overfitting, we also made the assumption to keep the initial value $V_I$ fixed to $V_{inj}$. As noted before [15], releasing this constraint leads to acceptable fits by either the exponential or logistic models (to the price of deteriorated identifiability). However, the estimated values of $V_I$ are in this case biologically inconsistent.

On the other hand, the Gompertz model demonstrated excellent goodness-of-fit in all the experimental systems that we investigated. This is in agreement with a large body of previous experimental and clinical research works using the Gompertz model to describe unaltered tumor growth in syngeneic [60, 6, 10, 57] and xenograft [61, 62] preclinical models, as well as human data [55, 13, 12, 8]. The poor performances of the logistic model compared to the

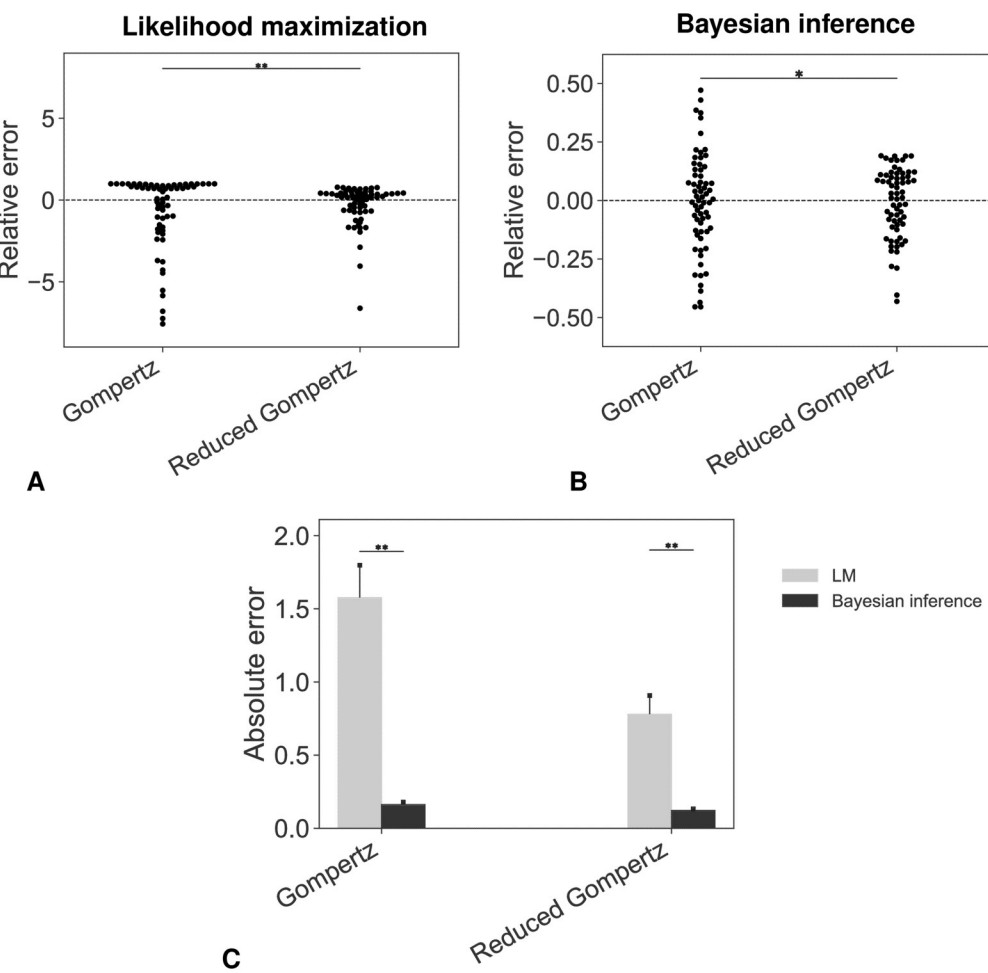

**Fig 5. Accuracy of the prediction models.** Swarmplots of relative errors obtained under likelihood maximization (A) or Bayesian inference (B) (* $p$-value < 0.05, ** $p$-value < 0.01, Levene's test). (C) Absolute errors: comparison between the different distributions (* $p$-value < 0.05, ** $p$-value < 0.01, Wilcoxon test). In (A) three extreme outliers were omitted (values of the relative error were greater than 20) for both the Gompertz and the reduced Gompertz in order to ensure readability. LM = Likelihood Maximization.

Gompertz model can be related to the structural properties of the models. The two sigmoid functions lie between two asymptotes ($V = 0$ and $V = K$) and are characterized by an initial period of fast growth followed by a phase of decreasing growth. These two phases are symmetrical in the logistic model, which is characterized by a decrease of the specific growth rate $\frac{1}{V}\frac{dV}{dt}$ at constant speed. On the other hand, the Gompertz model exhibits a faster decrease of the specific growth rate, at speed $-\frac{\beta}{V}$, or $e^{-\beta t}$ as a function of $t$, and the sigmoidal curve is not symmetric around its inflexion point. The logistic and Gompertz models belong to the same family of tumor growth equations and can be seen as specific cases of the generalized logistic model $\frac{dV}{dt} = \rho V\left(1 - \left(\frac{V}{K}\right)^\nu\right)$ [56, 15]. We also analyzed the latter model, which demonstrated good descriptive power but lacked robustness of convergence. Indeed, the SAEM algorithm converged to different estimates starting from different initial guesses of the parameters. This might be explained by the larger number of parameters (3) that led to identifiability problems. In addition, we found that values of $\nu$ able to describe the data were often very small ($< 10^{-3}$), thus suggesting convergence to the Gompertz model.

Similarly to previous reports [6, 11, 12, 13], we also found a very strong linear correlation between the two parameters of the Gompertz model, i.e. $\alpha$ the proliferation rate at injection and $\beta$ the rate of decrease of the specific growth rate. Importantly, this correlation is not due to a lack of identifiability of the parameters at the individual level, which we investigated and found to be excellent. Such finding motivated our choice to use a reduced Gompertz model, with only one individual-specific parameter, and one population-specific parameter. This model has been proposed before in the context of individual tumor growth curves [11, 51] but here we leveraged the population approach to ensure reliable estimation of the population-level parameter and statistical distribution of the individual-level parameter. Importantly, while previous studies had only investigated the resulting predictive power in only one animal [10] or using simulation data [51], here we rigorously demonstrated how the reduced Gompertz allows better backward (or forward, although not reported here) prediction of tumor size and time of initiation. This analysis was performed using state-of-the art techniques from predictive modeling (e.g. cross-validation), on a large number of animals.

The descriptive power of the reduced Gompertz model was found similar to the two-parameters Gompertz model. Critically, while previous work had demonstrated that two individual parameters were sufficient to describe tumor growth curves [15], these results now show that this number can be reduced to one. Interestingly, we found different values of the carrying capacity $K$ for the breast and the lung cancer cell lines measured by volume ($K = 2600$ mm$^3$ and 12300 mm$^3$, respectively), in contrast with previous claims [11]. This suggests that there might not be a characteristic saturation point within a species [51] but the carrying capacity could be a typical feature of a tumor type in an animal model. From (10), the population constant $k$ depends on the value of the parameter $V_{\text{inj}}$, therefore it cannot be viewed as a universal constant of tumor growth. However, it can be considered as a common trait within a species with similar characteristics (such as tumor type and value of $V_{\text{inj}}$). We used the formulations of the Gompertz (3) and reduced Gompertz (10) in order to define $\alpha$ as the specific growth rate at injection, which could be compared to the *in vitro* proliferation rate $\lambda$. This could be leveraged clinically to predict past or future tumor growth kinetics based on proliferation assays, derived from a patient's tumor sample.

The reduced Gompertz model, combined to Bayesian estimation from the population prior, allowed to reach good levels of accuracy and precision of the time elapsed between the injection of the tumor cells and late measurements, used as an experimental surrogate of the age of a given tumor. Importantly, performances obtained without using a prior were substantially worse. The method proposed herein remains to be extended to clinical data, although it will not be possible to have a firm confirmation since the natural history of neoplasms from their inception cannot be reported in a clinical setting. Nevertheless, the encouraging results obtained here could allow to give informative estimates, even if approximative. Importantly, the methods we developed also provide a measure of precision, which would give a quantitative assessment of the reliability of the predictions. For clinical translation, $V_{\text{inj}}$ should be replaced by the volume of one cell $V_c = 10^{-6}$ mm$^3$. Moreover, because the Gompertz model has a specific growth rate that tends to infinity when $V$ gets arbitrarily small, our results might have to be adapted with the Gomp-Exp model [63, 24].

Our methodology might face multiple challenges for future clinical applications. First, it is difficult to fully characterize unperturbed tumor kinetics in humans and only few studies support the evidence that it follows a gompertzian growth [8]. This is due to the limited number of available observations in the clinic and to the fact that saturation of human tumors is almost never reached, since it coincides with an advanced stage of the cancer where patients usually receive a treatment. Moreover, human tumor growth curves, even if limited to the same organ and histological type, exhibit a substantially larger variability than in *in vivo* experimental

settings where immortalized cancer cell lines are injected in genetically identical mice. Here, we have proven that a given animal model (i.e. same mice, tumor type and number of injected cells) is characterized by a common tumor growth constant, that defines the saturation point. In the human setting, it could be interesting to analyze this constant as a function of some covariates (such as weight, sex, tumor type). Eventually, in the Gompertz model we haven't considered that the initial phase of tumor growth might be affected by intrinsic stochasticity. Our choice was motivated by the large number of injected cells (of the order of $10^6$) that allowed us to consider the initial variability to be negligible. For accurate clinical translation, stochasticity should ideally be taken into account to model the initial stages of tumor growth.

Personalized estimations of the age of a given patient's tumor would yield important epidemiological insights and could also be informative for routine clinical practice [39]. By estimating the period at which the cancer initiated, it could give clues on the possible causes (environmental or behavioral) of neoplastic formation. Moreover, reconstruction of the natural history of the pre-diagnosis tumor growth might inform the presence and extent of invisible metastasis at diagnosis. Indeed, an older tumor has a greater probability of having already spread than a younger one. Altogether, the present findings could contribute to the development of personalized computational models of metastasis [24, 64, 65].

## Supporting information

**S1 Table. Statistical indices of the tumor growth models (lung, volume).** Models ranked in ascending order of AIC (Akaike information criterion). Other statistical indices are the log-likelihood estimate (-2LL) and the Bayesian information criterion (BIC).
(PDF)

**S2 Table. Statistical indices of the tumor growth models (breast, fluorescence).** Models ranked in ascending order of AIC (Akaike information criterion). Other statistical indices are the log-likelihood estimate (-2LL) and the Bayesian information criterion (BIC).
(PDF)

**S3 Table. Parameter values estimated with the SAEM algorithm (lung, volume).** Fixed effects (typical values) of the parameters of the different models. $\omega$ is the standard deviation of the random effects. $\sigma$ is vector of the residual error model parameters. Last column shows the relative standard errors (R.S.E.) of the estimates.
(PDF)

**S4 Table. Parameter values estimated with the SAEM algorithm (breast, fluorescence).** Fixed effects (typical values) of the parameters of the different models. $\omega$ is the standard deviation of the random effects. $\sigma$ is vector of the residual error model parameters. Last column shows the relative standard errors (R.S.E.) of the estimates.
(PDF)

**S1 Fig. Diagnostic plots from population analysis (lung, volume).** Population analysis of experimental tumor growth kinetics. A) Visual predictive checks assess goodness-of-fit for both structural dynamics and inter-animal variability by reporting model-predicted percentiles (together with confidence prediction intervals (P.I) in comparison to empirical ones. B) Prediction distributions. C) Individual weighted residuals (IWRES) with respect to time. D) Observations vs predictions Left: exponential, Center: logistic, Right: Gompertz models.
(TIF)

**S2 Fig. Diagnostic plots from population analysis (breast, fluorescence).** Population analysis of experimental tumor growth kinetics. A) Visual predictive checks assess goodness-of-fit

for both structural dynamics and inter-animal variability by reporting model-predicted percentiles (together with confidence prediction intervals (P.I) in comparison to empirical ones. B) Prediction distributions. C) Individual weighted residuals (IWRES) with respect to time. D) Observations vs predictions Left: exponential, Center: logistic, Right: Gompertz models.
(TIF)

**S3 Fig. Individual fits from population analysis (lung, volume).** Three representative examples of individual fits (animal A, animal B and animal C) computed with the population approach relative to the exponential (left), the logistic (center) and the Gompertz (right) models.
(TIF)

**S4 Fig. Individual fits from population analysis (breast, fluorescence).** Three representative examples of individual fits (animal A, animal B and animal C) computed with the population approach relative to the exponential (left), the logistic (center) and the Gompertz (right) models.
(TIF)

**S5 Fig. Correlation between the Gompertz parameters and diagnostic plots of the reduced Gompertz model with the population approach (lung, volume).** Correlation between the individual parameters of the Gompertz model (A) and results of the population analysis of the reduced Gompertz model: visual predictive check (B), scatter plots of the residuals (C), prediction distribution (D) and examples of individual fits (E).
(TIF)

**S6 Fig. Correlation between the Gompertz parameters and diagnostic plots of the reduced Gompertz model with the population approach (breast, fluorescence).** Correlation between the individual parameters of the Gompertz model (A) and results of the population analysis of the reduced Gompertz model: visual predictive check (B), scatter plots of the residuals (C), prediction distribution (D) and examples of individual fits (E).
(TIF)

**S7 Fig. Backward predictions computed with likelihood maximization (LM) and with Bayesian inference (lung, volume).** Three examples of backward predictions of individuals A, B and C computed with likelihood maximization (LM) and Bayesian inference: Gompertz model with likelihood maximization (first row); reduced Gompertz with likelihood maximization (second row); Gompertz with Bayesian inference (third row) and reduced Gompertz with Bayesian inference (fourth row). Only the last three points are considered to estimate the parameters. The grey area is the 95% prediction interval (P.I) and the dotted blue line is the median of the posterior predictive distribution. The red line is the predicted initiation time and the black vertical line the actual initiation time.
(TIF)

**S8 Fig. Error analysis of the predicted initiation time (lung, volume).** Accuracy of the prediction models. Swarmplots of relative errors obtained under likelihood maximization (A) or Bayesian inference (B). (C) Absolute errors: comparison between the different distributions ($^*$ $p$-value $< 0.05$, $^{**}$ $p$-value $< 0.01$).
(TIF)

**S9 Fig. Backward predictions computed with likelihood maximization (LM) and with Bayesian inference (breast, fluorescence).** Three examples of backward predictions of

individuals A, B and C computed with likelihood maximization (LM) and Bayesian inference: Gompertz model with likelihood maximization (first row); reduced Gompertz with likelihood maximization (second row); Gompertz with Bayesian inference (third row) and reduced Gompertz with Bayesian inference (fourth row). Only the last three points are considered to estimate the parameters. The grey area is the 95% prediction interval (P.I) and the dotted blue line is the median of the posterior predictive distribution. The red line is the predicted initiation time and the black vertical line the actual initiation time.
(TIF)

**S10 Fig. Error analysis of the predicted initiation time (breast, fluorescence).** Accuracy of the prediction models. Swarmplots of relative errors obtained under likelihood maximization (A) or Bayesian inference (B). (C) Absolute errors: comparison between the different distributions (* $p$-value $< 0.05$, ** $p$-value $< 0.01$).
(TIF)

## Author Contributions

**Conceptualization:** Sébastien Benzekry.

**Data curation:** Cristina Vaghi, Anne Rodallec, Raphaëlle Fanciullino, Joseph Ciccolini, Michalis Mastri, John M. L. Ebos.

**Formal analysis:** Cristina Vaghi, Jonathan P. Mochel, Clair Poignard, Sébastien Benzekry.

**Investigation:** Cristina Vaghi, Sébastien Benzekry.

**Methodology:** Cristina Vaghi, Sébastien Benzekry.

**Project administration:** Sébastien Benzekry.

**Resources:** Anne Rodallec, Raphaëlle Fanciullino, Joseph Ciccolini, Michalis Mastri, John M. L. Ebos.

**Software:** Cristina Vaghi.

**Supervision:** Sébastien Benzekry.

**Validation:** Sébastien Benzekry.

**Visualization:** Cristina Vaghi.

**Writing – original draft:** Cristina Vaghi, Sébastien Benzekry.

**Writing – review & editing:** Joseph Ciccolini, Jonathan P. Mochel, Sébastien Benzekry.

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
