## [Decision Letter · Decision Letter 0]

29 Aug 2019

Dear Dr Benzekry,

Thank you very much for submitting your manuscript 'A reduced Gompertz model for predicting tumor age using a population approach' for review by PLOS Computational Biology. Your manuscript has been fully evaluated by the PLOS Computational Biology editorial team and in this case also by three independent peer reviewers. While the reviewers found your work an interesting and well-designed implementation of data-driven modelling and model selection, they raised some substantial concerns about the manuscript as it currently stands. In particular, the validity of some of your conclusions, and the generality of others, is questioned by the reviewers. While your manuscript cannot be accepted in its present form, we urge you to consider a revised version in which the issues raised by the reviewers have been adequately addressed. We cannot, of course, promise publication at that time.

Sincerely,

Zvia Agur, PhD

Guest Editor

PLOS Computational Biology

Feilim Mac Gabhann

Editor-in-Chief

PLOS Computational Biology

[LINK]

Reviewer's Responses to Questions

**Comments to the Authors:**

Reviewer #1: Dear Professor Agur,

The manuscript by Benzekry and coworkers is very interesting and insightful but it needs some important improvements.

Thus, I suggest major revisions.

The following is a list of key points to be reworked:

1. I liked a lot the basic idea to use mixed models as a way to relate the single individual laws of growth population-wide data. However, I disagree with the choice of comparing the Gompertz model only with the simplest two other ODE models of tumor growth: the exponential and the classical logistic model. The comparison must be done also with respect to more realistic models such as: the generalized logistic (V’ = a V – b V^{n} with 0<n>

2. The fact that the classical logistic law V’= p V – q V^2 did not perform well is not so surprisingly. See for example some consideration at the end of the section 2 of d’Onofrio, Chaos, solitons and Fractals (2009): the classical logistic model might correspond to space-independent cellular interactions, which are non-physical.

3. The authors have detected a correlation between the parameters alpha and beta. If I well understood what the author mean, this correlation is not new but has been reported in literature in the seventies. See, for example, section 5.5 of the Wheldoin’s book (ref 40). This is not explicitly said in the manuscript: the authors simply cite references 11 and 26 and only in the discussion they list a series of previous similar results. The fact that this correlation is something previously known in literature must be explicitly written in the revised version where the correlation is described, not only in the discussion.

4. Are the authors sure that the reduced Gompertz model has not been defined previously ?

5. In non-oncological literature on biological growth there is a number of works that apply nonlinear mixed effects modeling to Gompertz model. They ought to be mentioned and reviewed in the introduction of the revised version of the manuscript.

6. There is also some works applying mixed-effects statistical modeling to investigate tumor growth described by simple ODEs models, including the Gompertz and other ode-based models (e.g the above mentioned paper by Ribba et al or the paper by Hartung et al cancer research (2014) and, of course, Benzekry et al, 2014). Indeed, currently the authors only generically wrote “However, to our knowledge, a detailed study of statistical properties of classical growth models at the level of the population (i.e. integrating structural dynamics with inter-animal variability) yet remains to be reported. Longitudinal data analysis with nonlinear mixed-effects is an ideal tool to perform such a task [17, 18].” Thus, the authors ought to clarify in detail which are the differences between their study and the previous ones (not limited to the above-mentioned papers….)

7. Summarizing points 3, 5 and 6: my feeling is that a substantial effort is needed to compare the present work to previous literature, explicitly stressing what is really new in this work (a lot) and what it has been previously published.

8. In my opinion, the description of the Bayesian inferences ought to be more detailed. This would help both those that are more oriented to the frequentist approach in Statistics, and those (e.g. biomathematicians) who are not expert at all in statistics

9. As far as the “tumor age” is concerned, I would be much more prudent due to the fact that tumour growth is both affected by intrinsic stochasticity (of course, especially in the initial stages of growth) and also extrinsic stochasticity. What the authors brilliantly computed is merely an estimation of the average tumor age, and the associated confidence-credibility intervals of that estimate. This is something radically different from inferring the probability density of the random variable “tumor age”, which can only be done in the framework of a stochastic tumour growth model

10. In the discussion the authors must add a detailed discussion of the limitations of their work.

Kind Regards,

A Referee

Reviewer #2: The manuscript considers an interesting problem of predicting tumor age. The main model used for the data analysis is the Gompertz model and its reduced form. This is correct from my point of view. The reduced version is well argued.

On the other hand, the authors compare the classic logistic model with constant carrying capacity (which should be understood as maximal tumor size here) with the form of Gompertz model in which carring capacity depends on initial data (as it is in the original Gompertz model). Typically, for the comparizon the Gompertz model is rescaled in such a way that there is a carying capacity independent of the initial value, and then the comparizon between the models makes sense. The authors should address this problem in their manuscript.

Moreover, in both type of the Gompertz model (full and reduced) there arises a problem of dependance on V_inj, which is known in vitro experiments but uknown in vivo. This is probably the main problem considering the utility of the proposed method.

I have also foud some editorial and language mistakes, like:

- lines 108/109 "number of injected" WHAT?

- line 116 "litterature" should be literature

- in Formula (6): what is p here? In line 168 p is the number of parameters!

- in line 158: "equation(7)" - lack of space.

Reviewer #3: Review is uploaded as an attachment</n>

**Have all data underlying the figures and results presented in the manuscript been provided?**

Reviewer #1: None

Reviewer #2: Yes

Reviewer #3: None

PLOS authors have the option to publish the peer review history of their article (what does this mean?). If published, this will include your full peer review and any attached files.

Reviewer #1: No

Reviewer #2: No

Reviewer #3: No

---

## [Editor Report · Decision Letter 1]

11 Dec 2019

Dear Dr Benzekry,

Thank you for submitting a revised manuscript 'A reduced Gompertz model for predicting tumor age using a population approach.' The paper is interesting and most of the reviewers' comments have been addressed satisfactorily, and the discrepancies still existing are minor. Nevertheless, we would ask you to make a minor revision before the manuscript is finally accepted for publication.

One reviewer still has reservations on the implications of the assumption that k may be constant. The reviewer trusts your findings that from the statistical point of view, this can be assumed here without loss of goodness of fit. However, he doubts your claim that this is a universal property. Simply put, in this model we have carrying capacity K = Vinj exp(k). If k is fixed for the tumor type and animal, this means that by repeating this experiment, and injecting say 0.1*Vinj one will obtain a 10 times smaller carrying capacity; or inversely, injecting 10 times more cells will allow tumor to grow asymptotically to a 10 times larger size. This dependence of saturation level on the initial size seems biologically unsound.

The reviewer does not think this invalidates your results, but that it calls for more careful interpretation. Possibly, the real "natural constant" is K, and the value of k would change if you inject larger number of the same cells into the same animals. This concern should be elaborated in the discussion, especially when you envisage the possible use of the model to trace the tumor growth back in time to the size of 1 cell.

The second concern the reviewer raises is the insufficient review of previous works on “mathematical models for tumor growth, which have been previously studied and compared at the level of individual kinetics and for prediction of future tumor growth” (line 53). Please note that there is previous work on the subject, e.g.,

Kronik N., Kogan Y., Elishmereni M., Halevi-Tobias K., Vuk Pavlović S., Agur Z. Predicting Effect of Prostate Cancer Immunotherapy by Personalized Mathematical Models PLoS One 2010 5(12) (and it deals with the clinical data) and several other publications in the last 10 years. Please, study the literature and include previous work on the subject appropriately in your review.

We would therefore like to ask you to modify the manuscript according to the review recommendations before we can consider your manuscript for acceptance. Your revisions should address the specific points made by each reviewer and we encourage you to respond to particular issues Please note while forming your response, if your article is accepted, you may have the opportunity to make the peer review history publicly available. The record will include editor decision letters (with reviews) and your responses to reviewer comments. If eligible, we will contact you to opt in or out.raised.

- Supporting Information uploaded as separate files, titled 'Dataset', 'Figure', 'Table', 'Text', 'Protocol', 'Audio', or 'Video'.

We hope to receive your revised manuscript within the next 30 days. If you anticipate any delay in its return, we ask that you let us know the expected resubmission date by email at ploscompbiol@plos.org.

Sincerely,

Zvia Agur, PhD

Guest Editor

PLOS Computational Biology

Feilim Mac Gabhann

Editor-in-Chief

PLOS Computational Biology

---

## [Editor Report · Decision Letter 2]

6 Jan 2020

Dear Dr Benzekry,

We are pleased to inform you that your manuscript 'A reduced Gompertz model for predicting tumor age using a population approach' has been provisionally accepted for publication in PLOS Computational Biology.

In the meantime, please log into Editorial Manager at https://www.editorialmanager.com/pcompbiol/, click the "Update My Information" link at the top of the page, and update your user information to ensure an efficient production and billing process.

One of the goals of PLOS is to make science accessible to educators and the public. PLOS staff issue occasional press releases and make early versions of PLOS Computational Biology articles available to science writers and journalists. PLOS staff also collaborate with Communication and Public Information Offices and would be happy to work with the relevant people at your institution or funding agency. If your institution or funding agency is interested in promoting your findings, please ask them to coordinate their releases with PLOS (contact ploscompbiol@plos.org).

Thank you again for supporting Open Access publishing. We look forward to publishing your paper in PLOS Computational Biology.

Sincerely,

Z Agur, PhD

Guest Editor

PLOS Computational Biology

Feilim Mac Gabhann

Editor-in-Chief

PLOS Computational Biology

---

## [Editor Report · Acceptance letter]

13 Feb 2020

PCOMPBIOL-D-19-00943R2 

Population modeling of tumor growth curves and the reduced Gompertz model improve prediction of the age of experimental tumors

Dear Dr Benzekry,

I am pleased to inform you that your manuscript has been formally accepted for publication in PLOS Computational Biology. Your manuscript is now with our production department and you will be notified of the publication date in due course.

With kind regards,

Matt Lyles
